# Fault Detection via 2.5D Transformer U-Net with Seismic Data Pre-Processing

**Zhanxin Tang [1], Bangyu Wu [1,*] , Weihua Wu [2] and Debo Ma [3]**

1. School of Mathematics and Statistics, Xi'an Jiaotong University, Xi'an 710049, China
2. School of Software Engineering, Xi'an Jiaotong University, Xi'an 710049, China
3. Research Institute of Petroleum Exploration & Development, PetroChina, Beijing 100083, China
* Correspondence: bangyuwu@xjtu.edu.cn

**Abstract:** Seismic fault structures are important for the detection and exploitation of hydrocarbon resources. Due to their development and popularity in the geophysical community, deep-learning-based fault detection methods have been proposed and achieved SOTA results. Due to the efficiency and benefits of full spatial information extraction, 3D convolutional neural networks (CNNs) are used widely to directly detect faults on seismic data volumes. However, using 3D data for training requires expensive computational resources and can be limited by hardware facilities. Although 2D CNN methods are less computationally intensive, they lead to the loss of correlation between seismic slices. To mitigate the aforementioned problems, we propose to predict a 2D fault section using multiple neighboring seismic profiles, that is, 2.5D fault detection. In CNNs, convolution layers mainly extract local information and pooling layers may disrupt the edge features in seismic data, which tend to cause fault discontinuities. To this end, we incorporate the Transformer module in U-net for feature extraction to enhance prediction continuity. To reduce the data discrepancies between synthetic and different real seismic datasets, we apply a seismic data standardization workflow to improve the prediction stability on real datasets. Netherlands F3 real data tests show that, when training on synthetic data labels, the proposed 2.5D Transformer U-net-based method predicts more subtle faults and faults with higher spatial continuity than the baseline full 3D U-net model.

**Keywords:** Transformer; 2.5D fault prediction; U-net; seismic data pre-processing

## 1. Introduction

Fault detection from seismic data is a critical component of hydrocarbon exploration and development workflow. Faults are control factors for reservoir delineation and fluid transportation, and can potentially create drilling hazards. Accurate fault mapping is a prerequisite for safe and efficient underground mining. Moreover, faults also play an important role in crust deformation, which is a key point for further analysis [1–4].

The primitive fault interpretation task is conducted manually, which relies heavily on the experience of experts. This approach can hardly meet the current production needs in terms of both accuracy and efficiency. With the rapid development of signal processing and computer technology, more and more data processing methods are being applied to assist fault interpretation. Various seismic data attributes based on anisotropy and coherence theories have been proposed, such as the gradient structure tensor, and variance and feature structure correlation [5–7]. Please refer to [8] for a general overview of seismic coherence for the delineation of structural and stratigraphic discontinuities. However, real seismic data inevitably contain noise and other data features that exhibit discontinuity characteristics similar to those of faults, which greatly affect the implementation of attribute-based fault interpretation methods.

In recent years, deep learning techniques have made tremendous advances and break-throughs in seismic fault detection. Fault interpretation is generally treated as an image

segmentation or classification task. Each pixel of the seismic image is identified by neural networks, with 1 indicating the presence of a fault and 0 indicating a non-fault. Early approaches consider fault detection mainly as a classification task. Xiong et al. [9] and Zhao et al. [10] classified each pixel point into 1 or 0 using windowed seismic slices as the input with the processing pixel location as the window center. Guitton [11] classified the central pixels in the 3D seismic data by entering the complete sub-volume. These methods predict each pixel individually, which is computationally intensive. Alternatively, more approaches treat fault detection as an image semantic segmentation task. Wu et al. [12] trained a 3D convolutional neural network, FaultSeg3D, using synthetic data and predicted faults on several field 3D datasets. An et al. [13] trained a neural network to perform end-to-end prediction on 2D seismic data. Wang et al. [14] ensembled the classification and segmentation CNNs based on knowledge distillation for seismic fault detection. In general, CNNs dominate existing deep learning fault detection approaches [8–22].

CNN is widely used in the field of image classification and computer vision tasks due to its effectiveness and efficiency. Since seismic data can also be treated as an image, the success of CNN applications benefits from the advantage of convolution feature extraction and assists in almost every subfield of geophysical tasks, such as denoising, missing trace reconstruction, and inversion [23–28]. However, the convolutional module operates data by extracting local features, which is not conducive to discovering long-range connections between pixels. Dosovitskiy et al. [29] demonstrated that the actual perceptual field of CNN is smaller than in theory, which does not facilitate the extraction of contextual global features for image classification. Locally, the faults exhibit a polarity change in the horizontal seismic waveform. In contrast, at a large scale, faults demonstrate geometric structural patterns, such as lines in 2D and surfaces in 3D. CNN methods based on local feature extraction overlook large scale connections and tend to generate faults with broken geometric structures. Dou et al. [15] pointed out that general CNN models, such as U-net (originally devised for medical images) and FCN (for natural images) based networks, lose high frequency information related to edges after multiple down sample pooling. This leads to poor identification of subtle and dense faults. Dou et al. [30] achieved improved results by reducing the down sampling layers and replacing pooling with convolution.

Transformer is an efficient module for contextual information extraction. It was first applied to natural language processing and, recently, has been widely used in image processing. Each point in the output feature map of the Transformer layer is calculated using all points of the input feature map. In comparison, each point in the output of the convolution layer uses only the information with the size of the convolution kernel in the input feature map. Transformer uses Self-Attention to capture global contextual information and establishes remote dependency on the seismic data, thus extracting full scale features. Dosovitskiy et al. [29] introduced Transformer to the image field called ViT and achieved improved results for image classification. The Transformer-based Segformer network proposed by Xie et al. [31] is successfully used for medical image segmentation. Liu et al. [32] proposed Swin Transformer, which improves the performance while greatly reduces the computation cost.

Using 3D seismic data to predict faults can make use of full spatial information, but requires more computational resources. The computation complexity of Transformer is quadratic with the size of the input feature map. Although window-based Self-Attention from Swin Transformer reduces computational complexity, making predictions on 3D data still requires more GPU memory than CNNs. On the other hand, fault detection using a 2D data profile tends to lead to a lack of continuity between seismic slices. The 2.5D methods can achieve a balance between resource consumption and prediction accuracy [17,33]. They introduce information from neighboring seismic slices to enhance the continuity and stability of the predictions. These methods greatly reduce hardware requirements and achieve prediction results close to those of 3D networks. Based on the characteristics of the fault detection task, we designed a 2.5D Transformer-based U-shaped network named Transformer U-net for fault detection. The approach uses five adjacent seismic slices as

five channels in the network to predict faults in the middle slice. It increases the number of channels in the first layer of the network, with marginal extra computation. We use Overlapped Patch Merging (OPM) in Segformer to replace the pooling layer in the U-net network for down sampling. Then, the Swin Transformer block in Liu et al. [32] is used to extract features instead of convolutional layers.

Deep learning approaches are data-driven methods and training data is critically important for the network performance. A sufficient number of fault tags for field seismic data are difficult to obtain. In this regard, many existing methods are trained on synthetic seismic data. However, the differences between synthetic data and field data result in weak generalization performance of networks. Some methods have been proposed to mitigate this problem. Alohali et al. [34] introduced noise from real seismic data to the synthetic data. Dou et al. [30] used sparse real seismic labels for training. Zhou et al. [35] used transfer learning to eliminate this discrepancy. Pham et al. [36] proposed seismic data augmentation using GANs. Zheng et al. [37] generated seismic images with cycle-consistent adversarial networks. Jing et al. [38] generated the synthetic samples using point spread function (PSF) based convolution to obtain realistic synthetic seismic images.

In this paper, to overcome the drawbacks of the CNN-based approach, we design a 2.5D Transformer U-net for fault detection. In order to reduce the discrepancy between synthetic and field data, we propose a data pre-processing workflow for data standardization. The rest of the paper is organized as follows. In Chapter 2, we present the basic modules of the network and the overall network structure. The data standardization and the 2.5D data structure for training/prediction are introduced in Chapter 3. In Chapter 4, we first predict faults on Netherlands F3 field data, and the results are compared with Wu's 3D FaultSeg3D model [12]. Then, we conduct several comparison experiments to verify the effectiveness of 2.5D data. In Chapter 5, we discuss the role of data standardization in improving model performance and generalization enhancement. Chapter 6 is the conclusion of this paper.

## 2. Network Structure

### 2.1. Shifted Window-Based Self-Attention

The core of the Transformer in ViT [29] is the Self-Attention layer. For the input image H × W × C, where H, W, and C are height, width, and channel number, respectively, each pixel point is a vector with dimension C. These vectors are arranged as an N × C matrix for the input of Transformer, where N = H × W. After layer normalization, it enters the Self-Attention module. The Self-Attention module accepts a key matrix $K \in \mathbb{R}^{N \times d_k}$, a query matrix $Q \in \mathbb{R}^{N \times d_k}$ and a value matrix $V \in \mathbb{R}^{N \times C}$ as its input and produces an output $O \in \mathbb{R}^{N \times C}$, where $d_k$ denotes feature vector size of the query and key. Specifically, the input matrix is mapped into K, Q, and V. Then, the inner product is derived by query matrix Q with key matrix K and the softmax function is performed to obtain the attention weights; this is subsequently multiplied with the value matrix V to obtain the distributed representation of the sequence weighted by attention. The equation is expressed as follows:

$$\text{Attention}(Q, K, V) = \text{softmax}\left(\frac{QK^T}{\sqrt{d_k}}\right) V. \tag{1}$$

Let $\text{Sim} = \text{softmax}\left(\frac{QK^T}{\sqrt{d_k}}\right)$ in the formula. It is an N × N matrix, which contains the similarity information of any two pixels in the feature map. Then, the output vector of any position $i$ is calculated as $O_i = \sum_{j=1}^{N} Sim_{ij}V_j$. In this way, $O_i$ contains global information in the input feature map. Self-Attention is computed multiple times for a feature map and the results are stitched together, known as multi-head Self-Attention (MSA).

One drawback of ViT is that the computational complexity is squared with the number of pixels. To reduce the computation, Swin Transformer divides the input images into non-overlapping windows, and then performs MSA inside different windows, which is

named window-based multi-head Self-Attention (W-MSA). The computational complexity of W-MSA is linearly related to the image size, which greatly reduces the computation complexity of the model. Although W-MSA can reduce the computational complexity, the lack of information exchange between non-overlapping windows actually removes the transformer's ability to construct relationships globally using Self-Attention. The shifted window partitioning configuration (SW-MSA) was developed to exchange information across windows. The diagram of the shifted window operation is shown in Figure 1. The input feature map is first calculated with a W-MSA layer. Then, the windows of the W-MSA layer are shifted, and the W-MSA calculation is performed again with the new windows, namely SW-MSA. The shifted operation bridges the windows of the preceding W-MSA layer, providing connections among windows that significantly enhance modeling power. The shifted-window-based Self-Attention strategy greatly reduces calculation complexity while ensuring global information capture.

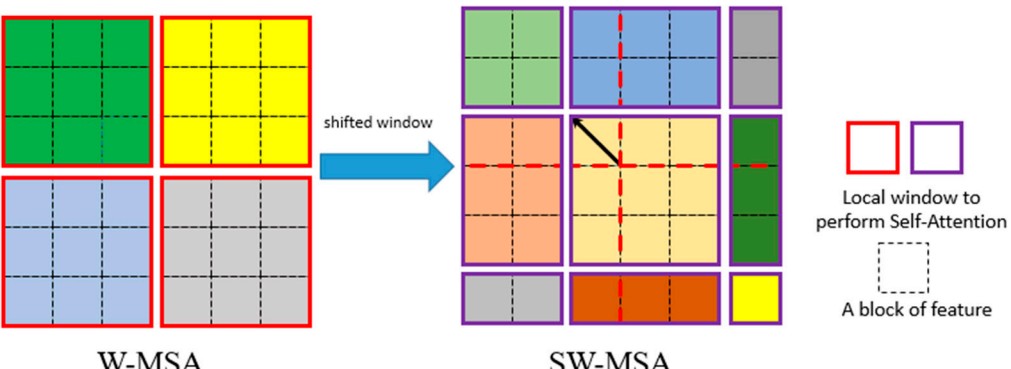

**Figure 1.** The shifted window operation in the Swin Transformer block. In the first layer, W-MSA, a regular window partitioning scheme is adopted, and Self-Attention is computed within each window. In the next layer, SW-MSA, the window partitioning is shifted, resulting in new windows. The Self-Attention computation in the new windows crosses the boundaries of the previous windows in the W-MSA layer, providing connections among windows (please refer to [32] for details).

*2.2. Swin Transformer Block for Feature Extraction*

As illustrated in Figure 2, a Swin Transformer block consists of W-MSA and SW-MSA modules, followed by two multi-layer perceptrons (MLPs) with Gaussian error linear unit (GELU) activation functions between them. A Layer Norm (LN) is applied before W-MSA, SW-MSA, and MLP, and a residual connection is applied to each module (as illustrated in Figure 2).

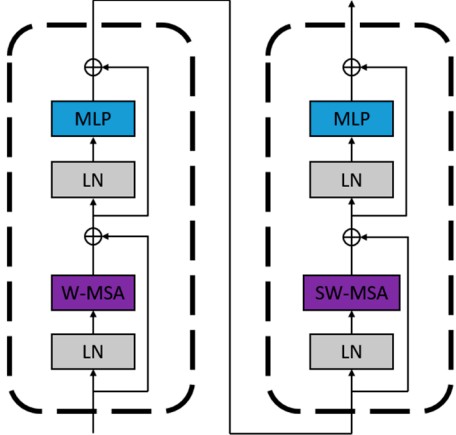

**Figure 2.** The architecture of a shifted window Transformer block. W-MSA and SW-MSA are multi-head Self-Attention modules with regular and shifted windowing configurations, respectively.

### 2.3. Overlapped Patch Merging (OPM) Down Sampling Layer

The OPM layer comes from Segformer, which learns position encoding implicitly through a block overlap strategy. The OPM layer has a similar function to that of the Patch Embedding (PBE) layer in ViT [29]. ViT chunks the input image without overlap and later maps the small P × P × 3 patches into a 1 × 1 × C vector. The non-overlapping chunks damage the continuity between blocks. OPM uses the overlap chunking to partition the feature map into several small patches with P × P overlapped small blocks. Then, features are fused while down sampling to obtain hierarchical features similar to convolutional neural networks. An example of OPM is shown in Figure 3, in which each 3 × 3 × 1 patch becomes a 1 × 1 × 3 vector after an OPM layer. This layer replaces the pooling layer in U-net. For our experiment, compared with PBE, using OPM retains precise location information, which can avoid spatial misalignment of faults.

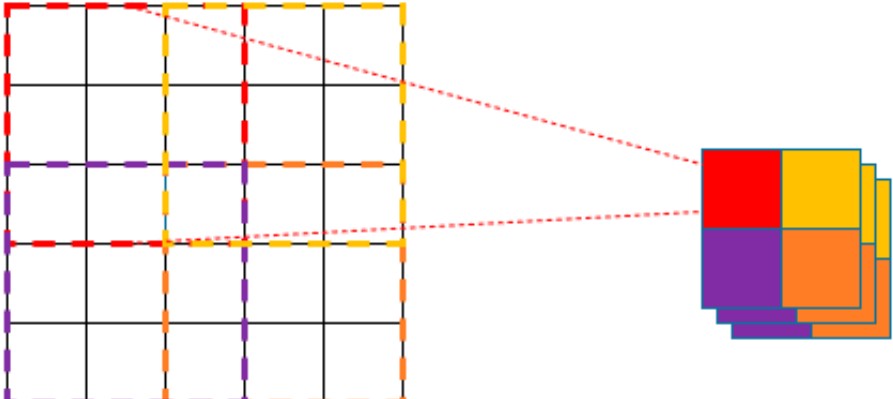

**Figure 3.** Schematic illustration of Overlapped Patch Merging (OPM).

### 2.4. Transformer U-Net for Seismic Fault Detection

The proposed overall structural framework of Transformer U-net is shown in Figure 4. The network has an encoder–decoder structure. The encoder has four stages. Each stage consists of one OPM down sampling and n Swin Transformer blocks. The number of Swin Transformer blocks, n, and the window size, w, are denoted as (n, w) above the Swin Transformer block. For the above parameter settings, we refer to Liu et al. [32]. The patch size in OPM blocks is 3 × 3. After an OPM block, the information of each pixel in the feature map comes from the nine pixels in the previous feature map. Therefore, the size of the feature map channels after the first down sampling increases 18 times. The feature map is half of the input after each down sampling block. For the encoder, the number of feature map channels in each stage is [18,36,72,144].

The decoder includes four up sampling stages and each stage is linked to the encoder by jump connections. The output layer of the network consists of a 3 × 3 and a 1 × 1 convolutional layer.

The proposed 2.5D Transformer U-net was trained on one 2080ti GPU with 50 epochs and batch size of 10. This requires about 5G of GPU memory consumption. It takes about 60 s to train one epoch. We used the Adam optimizer, and set the learning rate to 0.001.

### 2.5. Loss Function

Typically, fault detection in seismic data can be taken as an image segmentation task. However, there is a significant imbalance between positive (fault) and negative (non-fault) pixels, since faults are present in only a small fraction of the total data volume. Therefore, the use of a binary cross-entropy loss function causes the network to be biased towards predicting non-fault results.

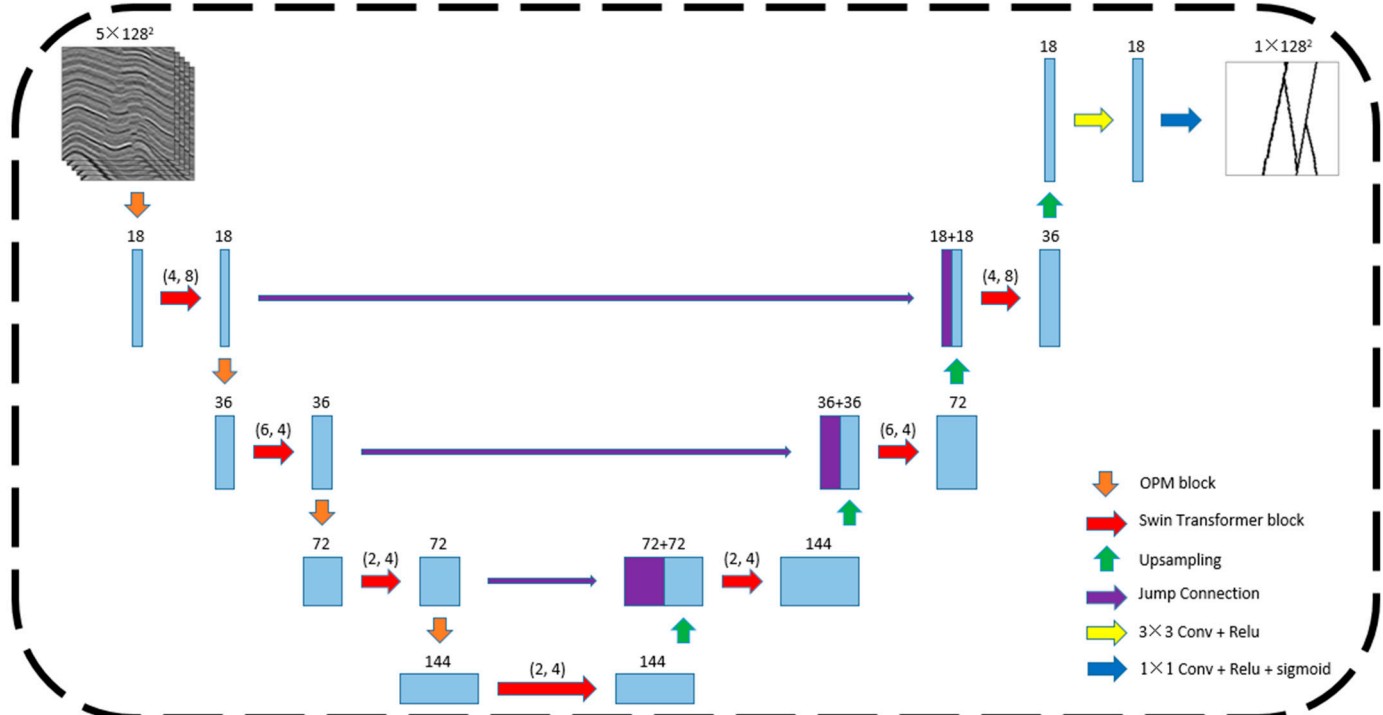

**Figure 4.** The proposed architecture of the 2.5D Transformer U-net for seismic fault interpretation. The number of Swin Transformer blocks, n, and the window size, w, is denoted as (n, w) above the Swin Transformer block.

In this paper, we use the balanced binary cross-entropy loss function proposed by Xie et al. [39], which is widely used for binary image classification tasks with unbalanced proportions of positive and negative samples:

$$L = -\beta \sum_{i=0}^{N} y_i \log(p_i) - (1-\beta) \sum_{i=0}^{N} (1-y_i) \log(1-p_i), \tag{2}$$

where $N$ denotes the number of pixels in the input seismic image, $y_i$ the binary label of the pixel, $p_i$ is the prediction probability of the neural network, and $\beta = \frac{\sum_{i=1}^{N}(1-y_i)}{N}$ is the ratio between non-fault pixels and total image pixels in the input seismic image.

## 3. Data Processing

### 3.1. Data Standardization

It is desirable for deep learning seismic fault detection methods to train on synthetic data and directly predict field seismic data. The difference between datasets may result in the model working well on the synthetic data but unsatisfactorily on the field data. The data standardization aims to reduce the differences between datasets to increase the generalization ability of networks.

From a physical point of view, the differences are manifested in spectrum [40], amplitude, noise [41], etc. Mathematically, this is mainly expressed as a difference between the data distribution [42], which is also known as the domain gap. The domain gap causes severe degradation in model generalization [43]. One commonly used standardization is z-score normalization, which is used in Wu et al. [12]:

$$x_{z-score} = \frac{x - \mu}{\delta} \tag{3}$$

where $x_{z-score}$ represents the data after z-score normalization, x represents the original data, $\mu$ and $\delta$ denote the mean and standard deviation of x. Z-score does not unify the data distribution. Figure 5a,b shows the amplitude distribution of one synthetic seismic data

sample in Wu et al. [12] and the field F3 dataset after z-score normalization. It can be found that, in terms of amplitude distribution, the kurtosis of the F3 data distribution is significantly larger than that of the synthetic data. The difference leads to poor generalization for the network when training on this synthetic data and applying it directly to the field data.

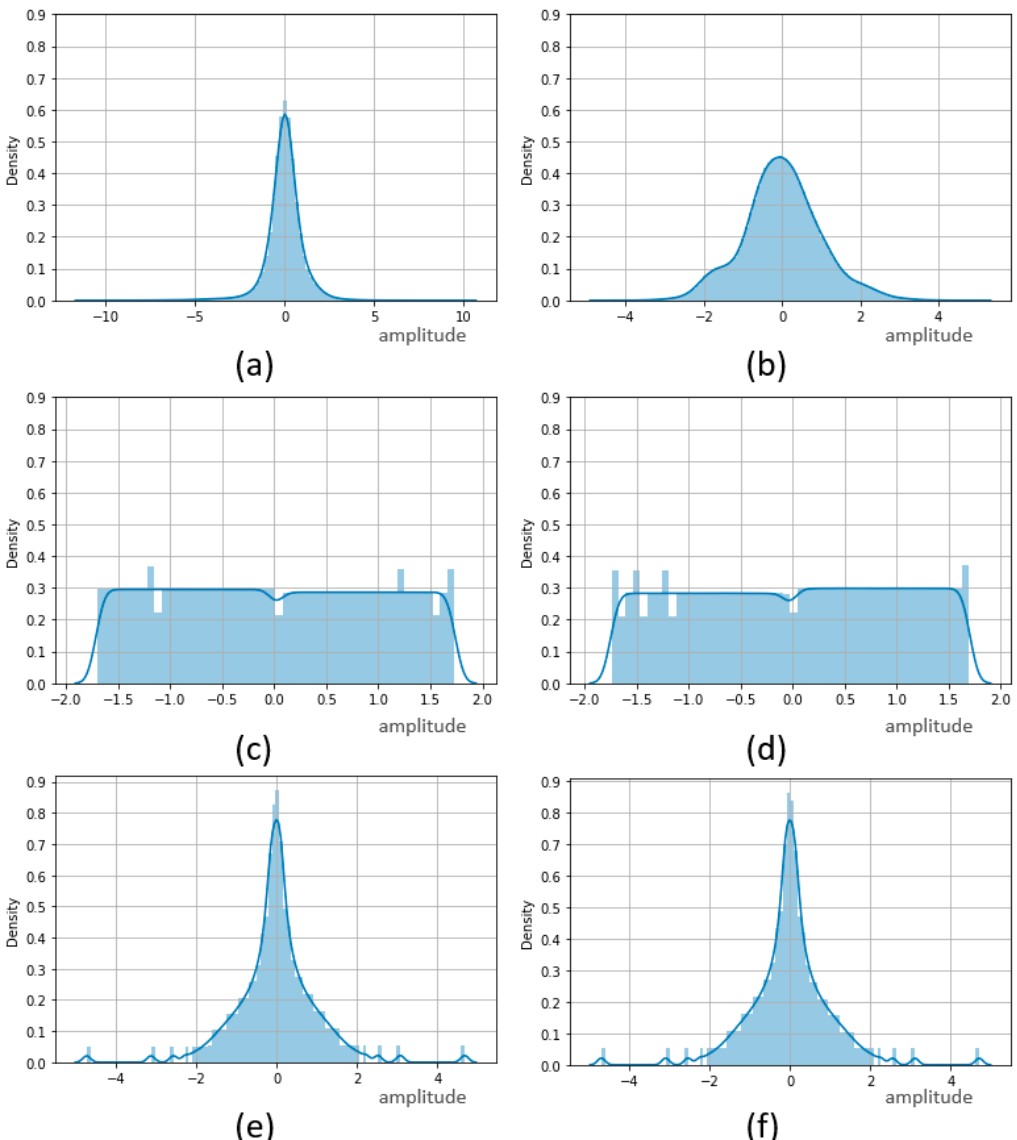

**Figure 5.** The amplitude distribution before standardization (**a**,**b**), after Step 1 with a z-score normalization (**c**,**d**), and after standardization (**e**,**f**) from the synthetic dataset (**b**,**d**,**f**) and the F3 dataset (**a**,**c**,**e**).

An ideal function maps the datasets to the same amplitude distribution while leaving the edge information unaffected, which can be expressed as follows:

$$D^* = f(D) \tag{4}$$

where D represents the original data, $D^*$ represents the data after standardization. $f(\cdot)$ is the function that maps $D^*$ from different datasets to a same distribution. Inspired by the standardization for continuous features in Cheng et al. [44], the data standardization we proposed is divided into two steps:

Step 1: Equal Frequency Normalization

Seismic data are different from natural images. We generally identify faults based on abrupt changes in reflected waves in seismic images. Only the relative amplitude information is needed for the fault identification task.

First, we normalize the original data using z-score normalization. Equal frequency normalization is based on percentile. We then normalize the data according to the position of each pixel value in the cumulative distribution function for the data body. The data are divided into buckets at equal frequencies to ensure that the pixel value in each bucket is approximately equal. Assume that there are total of n buckets, and the pixel x belongs to the *b*-th (b $\in \{1, 2, \cdots n\}$) bucket; then, the pixel value of $X_i$ after normalization is *b*.

For example, we divide the value of a particular data body into n = 100 intervals; the formula of Equal Frequency Normalization is shown below:

$$X = f_{z-score}(X_{source});$$

$$X_i^* = a; \ a \in \{1, 2, \cdots 100\} \text{ and } f_p(a-1) < X_i < f_p(a), \tag{5}$$

where $X_{source}$ represents the original data, $f_{z-score}(\cdot)$ is the z-score normalization function, $X_i$ is the value of the *i*-th point in *X*, $X_i^*$ is the value of the *i*-th point after processing, $f_p(\cdot)$ is the percentile function of the distribution function of this seismic data.

This method ensures that the data from different distributions can be mapped to an approximately uniform distribution. The positive and negative values of seismic data represent the positive and negative polarity, which is important information for judging faults. Therefore, we normalize the positive and negative values separately according to their absolute values. In addition, we ensure that their polarity remains unchanged. In practice, to mitigate the effect of noise, we first set a threshold $\varepsilon$, setting the pixel values within $-\varepsilon$ to $\varepsilon$ in the z-score normalized seismic image to 0. After that, we use the Equal Frequency Normalization for positive and negative values. The process does not change the positive and negative properties of the original data, and the final data take values in the range of $\{-100, -99, \cdots, 99, 100\}$.

The grayscale image can have 256 different values, while our normalized data have 201 values. We set 100 intervals because this not only facilitates the calculation, but also ensures that data after standardization provides enough information for fault interpretation tasks.

Step 2: Data Reconstruction

Two datasets with different distributions can be transformed into the same distribution using Step 1 only. However, there are some problems with this normalized data that will affect the effectiveness of the model training. From Figure 5a,b, we can see that the amplitude distribution density is higher the closer it is to zero in the original seismic image. Theoretically, for the homogeneous portion of the seismic image, no seismic signal is presented. However, due to the presence of noise, the actual amplitude is generally not 0, but a relatively small value. After Step 1, the data become uniformly distributed. The amplitude distribution density becomes the same. This operation amplifies small values compared to the original data. Figure 5c,d shows the distribution after Step 1 with a z-score normalization. From our experimental results, using only Step 1 results in regions with a lower signal-to-noise ratio being more easily identified as faults. To suppress the noise, we introduce operations to reconstruct the data after Step 1. The distribution curve after Data Reconstruction is shaped into a bell-shaped curve similar to Figure 5a.

In this case, suppressing noise means reducing smaller values or increasing larger values in the data after Step 1. A function $f_{DR}(x)$ needs to be designed. Its first and second derivative are greater than 0, and the output $f_{DR}(x) > 0$ when the input is positive. Finally, we use a z-score normalization to constrain the range of data values.

$$X_i^{**} = f_{z-score}(f_{DR}(X_i^*)), \tag{6}$$

where $X_i^{**}$ is the result after Data Reconstruction, $X_i^*$ is the result after Step 1.

Regarding the choice of $f_{DR}(\cdot)$, we tested several functions. Theoretically, all functions that satisfy the above conditions can be used in Step 2. However, improper function design may damage fault detection performance. For example, we compared two simple functions $f_{DR}^1(x) = x^3$ and $f_{DR}^2(x) = x^{1.1}$. Figure 6b,e shows the distribution and seismic image on F3 data using $f_{DR}^1$. Excessive suppression of small values using $f_{DR}^1$ loses seismic image information, which increases the discontinuity in prediction faults. Conversely, insufficient suppression using $f_{DR}^2$ leads to noise easily being predicted as faults, as shown in Figure 6c,f. In practice, we suggest choosing the one that minimizes the 2-Norm of the difference between the field data before and after standardization.

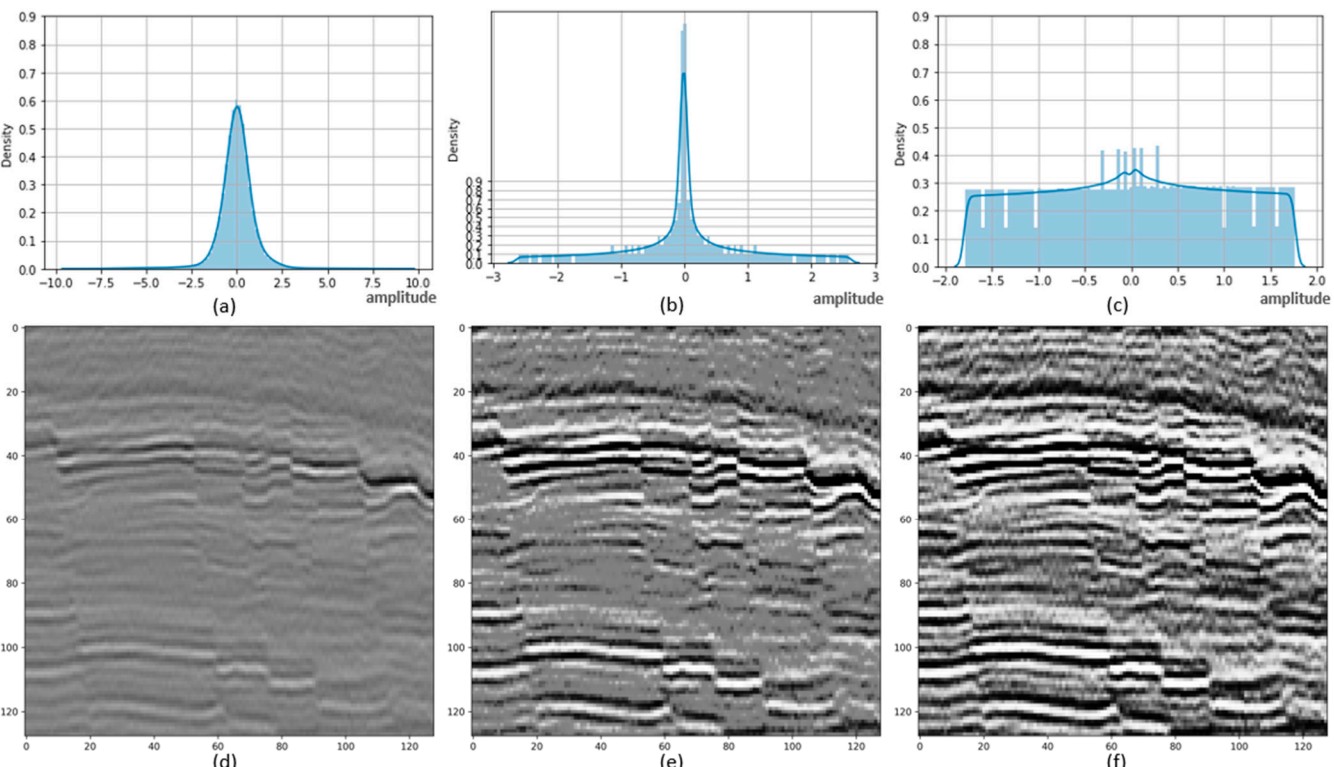

**Figure 6.** F3 data amplitude distributions (**a**–**c**) and corresponding seismic images (**d**–**f**). (**a**,**d**) original data; (**b**,**e**) standardized data with $f_{DR}(x) = x^3$; (**c**,**f**) standardized data with $f_{DR}(x) = x^{1.1}$.

In this paper, the function $f_{DR}(\cdot)$ is empirically set to:

$$\alpha \cdot \left(1 + \frac{3x}{100}\right)^3 + (1 - \alpha) \cdot \tan\left(\frac{x-1}{100} \cdot \frac{\pi}{2}\right), \ 0 < \alpha < 1$$

and we set $\alpha = \frac{1}{3}$. Figure 5e,f shows the distribution of the two datasets after standardization. At this point, both distributions from synthetic and F3 data become the same.

The role of data standardization is illustrated in Figure 5. Figure 5a,b shows the difference in amplitude distribution. This difference causes the model to perform worse on field data. After Step 1, as shown in Figure 5c,d, the distribution of the synthetic data becomes the same as that of the field data. However, the data tend to be uniformly distributed, which is not consistent with the actual seismic data. After Step 2, as shown in Figure 5e,f, while the field and synthetic data maintain the same distribution, the distribution curve becomes a bell-shaped curve that matches that of the actual data. Figure 7 shows the seismic data slices before and after standardization. It can be seen that standardization basically preserves the fault information. Figure 8 shows trace seismograms from both datasets before and after standardization. Compared to the original trace seismogram in Figure 8a,d, the amplitude is enhanced after Step 1, as shown in Figure 8b,e. This is consistent with

the analysis above, showing that Step 1 boosts the small values. After standardization, the overall waveform is almost indistinguishable from that before standardization, while the different datasets have the same distributions, as shown in Figure 8c,f.

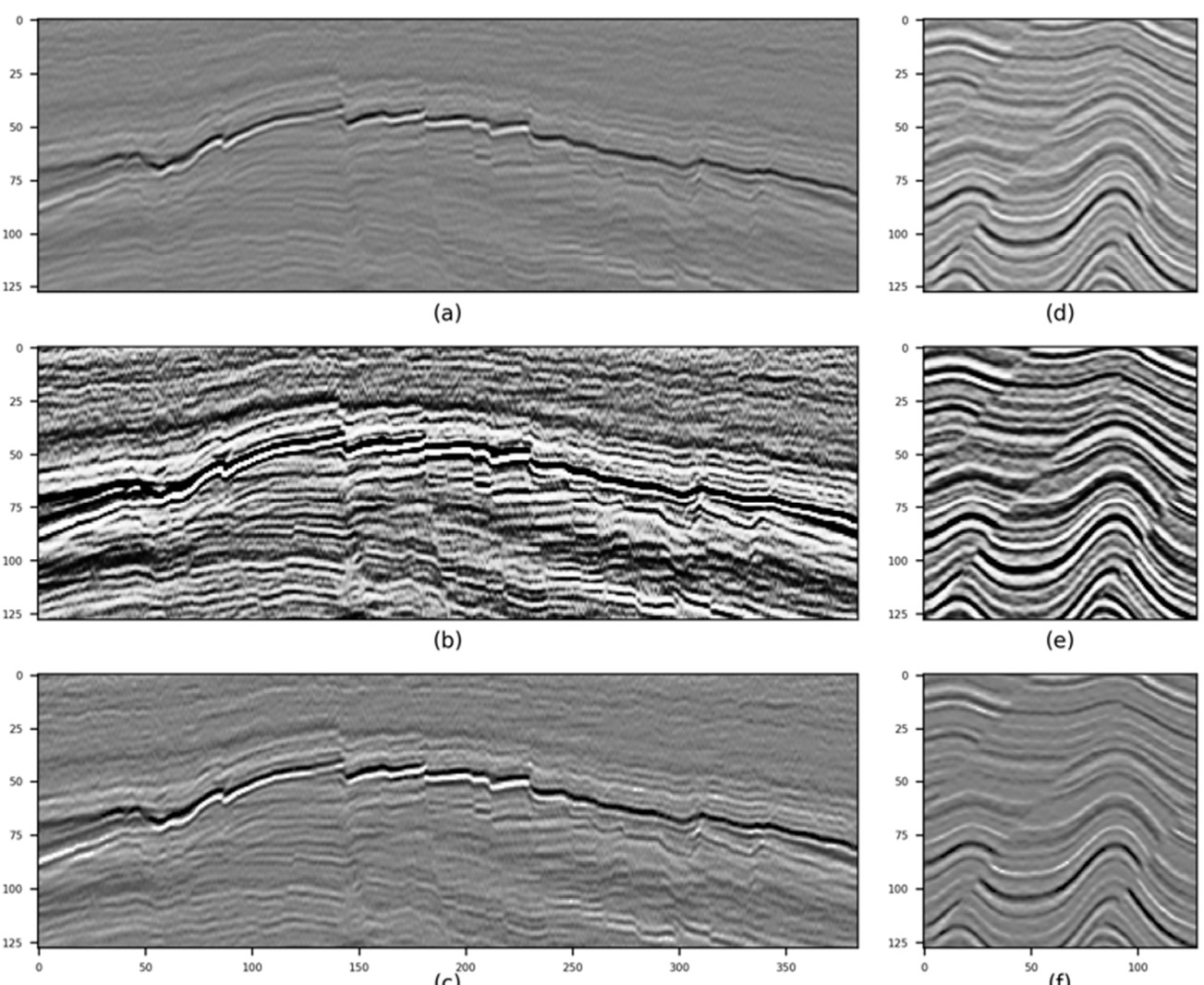

**Figure 7.** Seismic data before standardization (**a**,**d**), after Step 1 with a z-score normalization (**b**,**e**) and after standardization (**c**,**f**) from F3 (**a**–**c**) and synthetic (**d**–**f**) seismic datasets.

The proposed preprocessing is similar to Normalizing Flows (NF) [45], which is a general class of method that transforms an arbitrary data distribution to a simple underlying distribution by constructing a reversible transformation. The difference is that treatment of NF relates to the specific distributions, and is difficult to implement. The proposed method in this paper is irreversible, and is simple to operate.

### 3.2. 2.5D Data Assemble

For Transformer in image processing, the input of each pixel point is represented by a vector of dimension C. This is used as the input for each channel and for the Self-Attention calculation. The seismic data are 3D and the number of channels is 1. A fault is a three-dimensional structure, and whether each pixel point on a seismic slice is a fault or not is related to all other pixel points in the surrounding three-dimensional space. Specifically, for each seismic slice, we intercept n adjacent slices before and after to form a feature map with channel number of 2n+1, that is, 2.5D data as the input. The output is then the fault prediction of the center slice. In this paper, we set n to 2 based on numerical experiments.

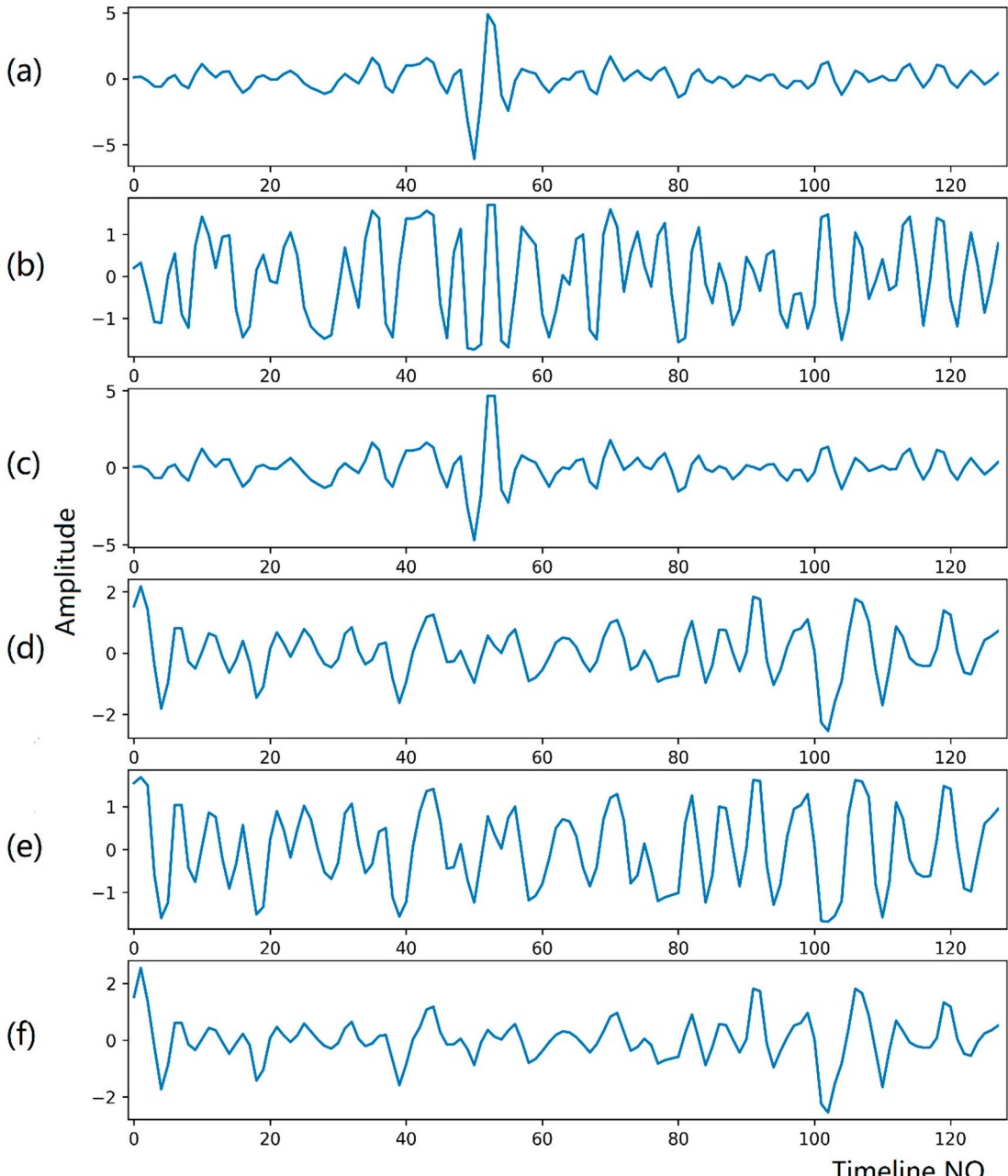

**Figure 8.** Trace seismogram from F3 (**a**–**c**) and synthetic seismic datasets (**d**–**f**) before standardization (**a**,**d**), after Step 1 with a z-score normalization (**b**,**e**) and after standardization (**c**,**f**).

### 3.3. Data Preprocessing Workflow

The proposed network was trained on the synthetic seismic dataset provided in Wu et al. [12]. Its training and validation sets contain 200 and 20 3D seismic data with corresponding labels, respectively. The size of each sample is 128 × 128 × 128. Each seismic sample is first preprocessed according to our proposed data standardization. After that, the samples are sliced into 2.5D data of $128^2 \times 5$ from both inline and crossline directions. The labels are taken from the third section, namely, the middle section. Since the 2D data are very similar on adjacent slices, according to Smith et al. [16], Dou et al. [46], and Zhu et al. [47], we select one from every 10 slices in the training set. After that, the processed data are used for training/validation/testing.

## 4. Experimental Results and Analysis

### 4.1. 2.5D Transformer U-Net Result on F3 Field Data

We tested the fault detection performance of 2.5D Transformer U-net on a portion of field Netherlands F3 seismic data. For prediction, we first preprocessed the F3 data volume with a size of $512 \times 384 \times 128$ in same way as for the training data. Then, the trained neural network was used to predict the F3 data from inline and crossline directions. The final prediction results were obtained by fusing the results in both directions. The formula is as follows:

$$P_i = \max\left( P_i^{IL}, P_i^{XL} \right) \tag{7}$$

where $P_i$ is the probability of the fault at the i-th point, $P_i^{IL}$ and $P_i^{XL}$ are fault probabilities predicted from inline and crossline directions, respectively.

The predicted faults are shown in Figure 9. Figure 9a,b shows the original and seismic data after standardization, respectively. Figure 9c shows the FaultSeg3D result provided by Wu et al. [12] for comparison. Figure 9d shows the proposed 2.5D Transformer U-net result. Although our method is 2.5D, the predicted fault results are improved in general. Comparing the faults indicated by the red arrows, the 2.5D Transformer U-net prediction has more complete fault junctions and superior continuity while FaultSeg3D predicts more broken structures.

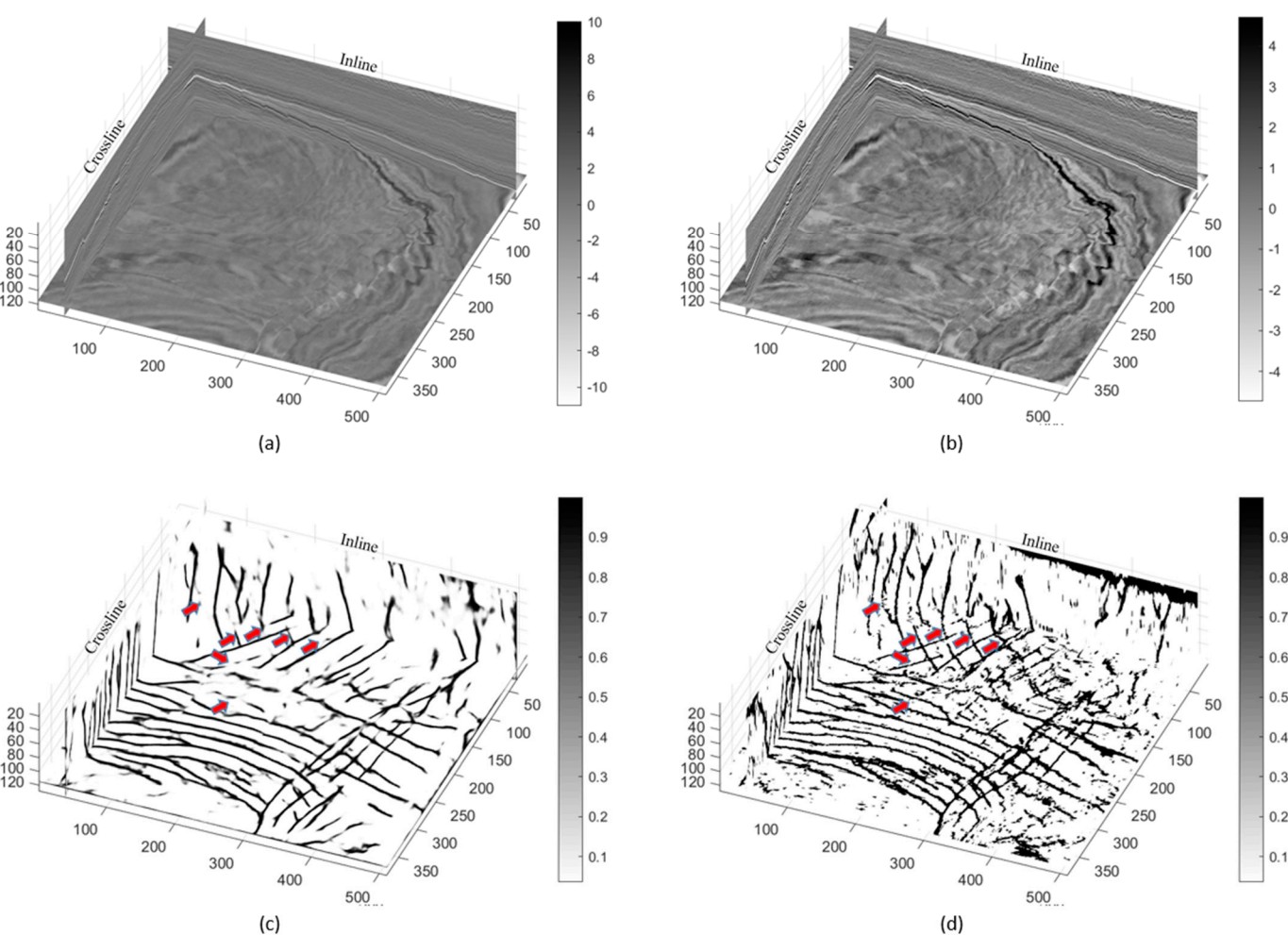

**Figure 9.** The F3 seismic data fault prediction results. (**a**) Original F3 data. (**b**) F3 data after standardization. (**c**) FaultSeg3D [12]. (**d**) Proposed 2.5D Transformer U-net.

*4.2. Comparison with Different Number of Channels*

First, we trained the proposed Transformer U-net using 2D (number of channels was 1) and 2.5D data. Figure 10 shows a time slice of the prediction results in different cases. Figure 10a–c shows the prediction results using 2.5D data. Figure 10d–f shows the prediction results for 2D data. Figure 10a,d shows predictions in the inline direction, which means predicting every inline slice then stitching them to a 3D result; Figure 10b,e shows predictions in the crossline direction; Figure 10c,f shows the results after fusion of the results from different directions by Equation (7).

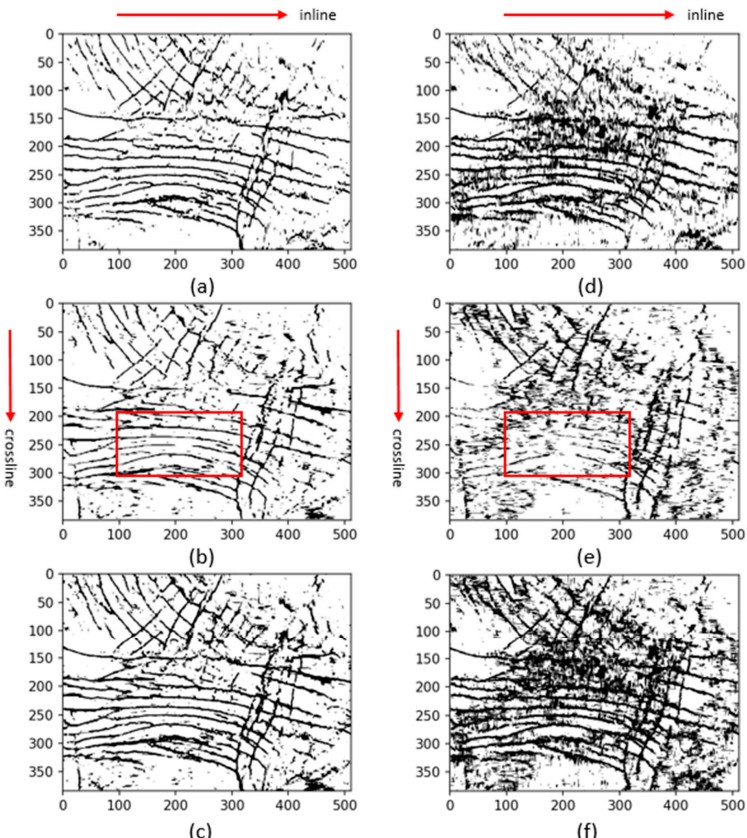

**Figure 10.** The time slices of prediction results for networks trained using 2.5D (**a–c**) and 2D (**d–f**) data. (**a**,**d**) are prediction results from the inline direction. (**b**,**e**) are prediction results from the crossline direction. (**c**,**f**) are results after fusion. The red arrows indicate the predicted direction, which is perpendicular to the 2D slices.

From the prediction results along the crossline direction, the fault in the red box in Figure 10b,e is almost parallel to the slice for prediction. It is extremely difficult to predict such faults from a single 2D slice. Dou et al. [30] illustrated that, the smaller the angle between the fault and the slice, the worse the prediction results. From the experimental results, it can be seen that such faults are completely unrecognizable when using 2D data, while they are successfully predicted using 2.5D data. This indicates that adding information before and after the slice is beneficial for 2D fault detection. The final results are shown in Figure 10c,f. The prediction results using 2D data are poor without coherent structures in the complex fault structure region, whereas 2.5D data significantly improve the prediction consistency.

Next, we explored the effect of 2.5D data with different numbers of channels. We conducted experiments using data with channel numbers 3, 5, and 7. Figure 11 shows their results on one time slice. It can be seen that the number of channels increases from 3 to 5 with some improvement in the prediction results and a decrease in the number of noise points. After increasing the number of channels to 7, the continuity improved slightly. At

the same time, the number of noise points increased. Adding more channels is not effective after the channel number exceeds a certain point. Too much information at a far distance interferes with the prediction results. The number of 2.5D data channels is generally not recommended to exceed 7.

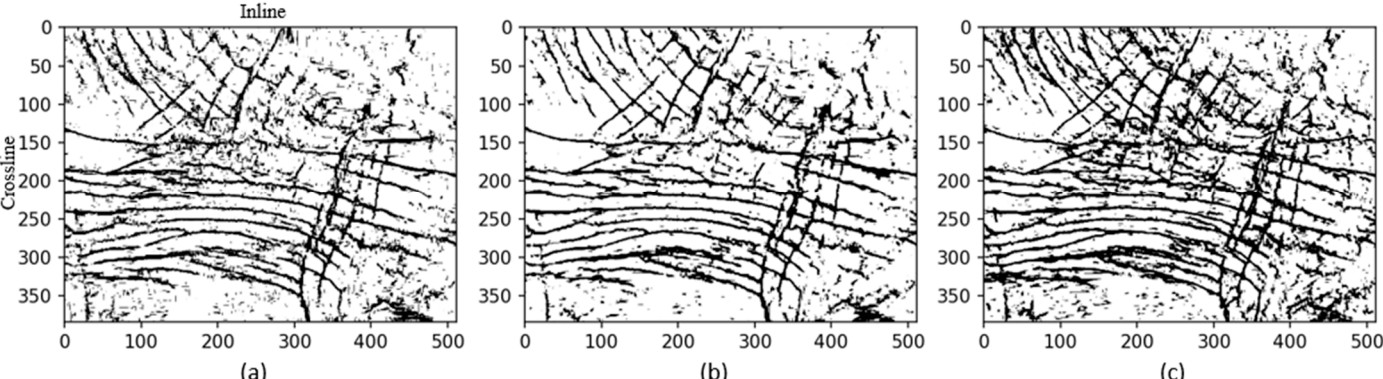

**Figure 11.** Prediction results using 2.5D data with different numbers of channels. (**a**) 3. (**b**) 5. (**c**) 7.

*4.3. Comparison Results with Different Down Sampling Layer*

We investigated the influence of different down sampling modules, OPM vs. PBE, on the model performance. The results are shown in Figure 12. Compared with OPM (Figure 12b), the fault predicted by 2.5D Transformer U-net with the PBE (Figure 12a) down sampling layer loses continuity between slices.

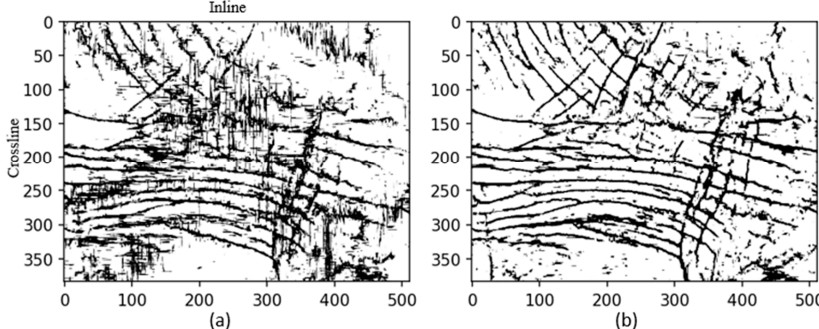

**Figure 12.** Results from 2.5D Transformer U-net with PBE (**a**) and OPM (**b**) down sampling layers, respectively.

In theory, PBE in ViT [29] is proposed for image classification tasks, while OPM in Segformer [31] is used for image segmentation. Image classification does not focus on pixel-level location information, as it only needs to classify the image as a whole. Image segmentation is more demanding and can be considered as a pixel-level classification. Compared to PBE, OPM extracts precise location information through the block overlap strategy. In this paper, fault detection is considered as an image segmentation task. Therefore, OPM is more suitable to this task, as demonstrated by the experimental results.

## 5. Discussion

Our 2.5D Transformer U-net approach achieved improved results compared with FaultSeg3D on the F3 field seismic dataset. To examine the role of data standardization in fault detection, we conducted the following investigation.

Generally, the training epoch with the smallest loss in the validation set is selected as the trained neural network. However, when training using synthetic data, the domain gap invalidates this rule. The better the model performs on synthetic seismic data, the worse it may perform on the field data. The severe overfitting phenomenon occurs. Overfitting

degrades the model's performance and stability. In addition, the selection of model parameters becomes difficult in this case. Users often need to make predictions on field data epoch by epoch, and then select the best one manually.

First, to verify that the standardization indeed improves the generalization performance of the network, we compared the faults predicted by networks trained with the original data and the standardized data. Figure 13 shows the performance of the network on fault detection in epochs 5, 8, 11, and 15. These epochs have decreasing losses on the validation set. Figure 13a–d shows results when trained on standardized synthetic data. Figure 13e–h shows results when trained on original synthetic data. The 15th epoch has the lowest validation loss. However, it can be seen in the figure that the network trained on original data has better results at epochs 5 and 8. After epoch 8, the performance decreases rapidly as the number of training epochs increases. In contrast, the network trained with standardized data tends to be stable. It does not show overfitting as serious as that of the original data.

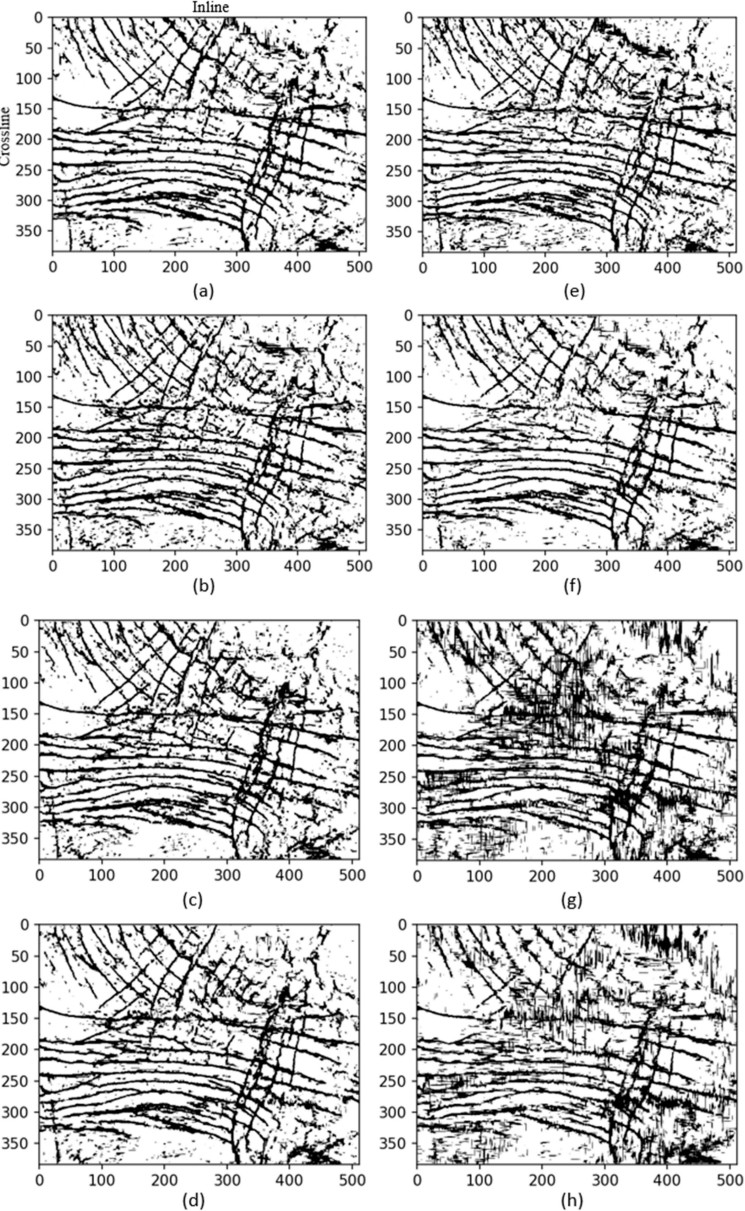

**Figure 13.** The Transformer U-net results with (**a**–**d**) and without (**e**–**h**) standardization on different epochs. (**a**,**e**) 5. (**b**,**f**) 8. (**c**,**g**) 10. (**d**,**h**) 15.

Then, we experimented with our data standardization on FaultSeg3D to verify its effectiveness in general. Since the code provided by the original authors is based on an older version of Tensorflow, we replicated FaultSeg3D on Pytorch as intractable compatibility issues occurred when run directly. Figure 14 shows the prediction results in training epochs 7, 15, 25, and 35. Their losses on the validation set also decrease. Figure 14a–d shows the results when trained on standardized synthetic data. Figure 14e–h shows the results when trained on original synthetic data. The results indicate that overfitting also occurs on FaultSeg3D using data without the proposed standardization. Compared with the network trained on original data, the network trained on standardized data results in stable and improved fault prediction.

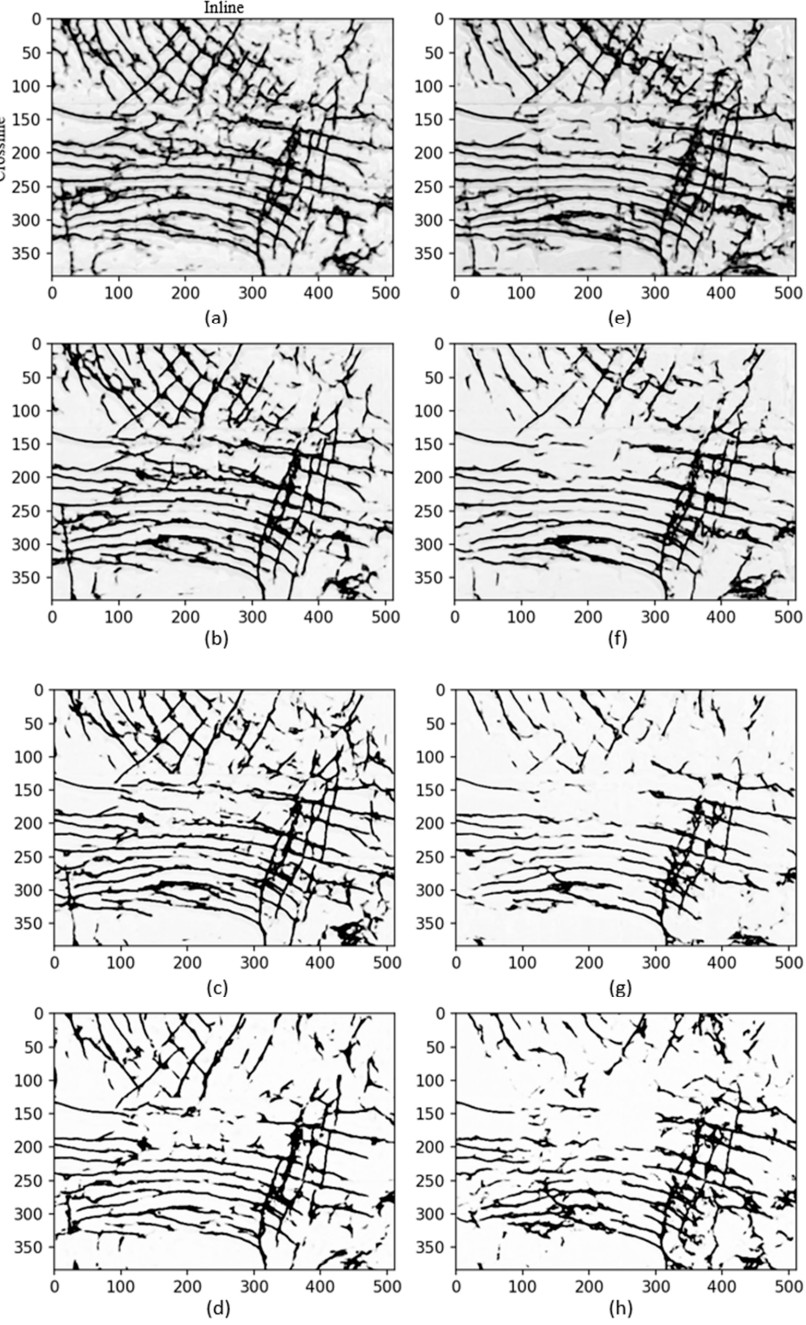

**Figure 14.** The FaultSeg3D results with (**a**–**d**) and without (**e**–**h**) standardization on different epochs. (**a**,**e**) 7. (**b**,**f**) 15. (**c**,**g**) 25. (**d**,**h**) 35.

In Figure 15a we show the 3D prediction result of FaultSeg3D using standardized data. Since the FaultSeg3D results we reproduce are somewhat different from Wu's [12], we list the prediction results provided by Wu et al. [12] in Figure 15b. Figure 15c,d shows the time slices in Figure 15a,b for a detailed comparison. Corresponding seismic data are shown in Figure 9a,b. The network trained on standardized data shows a significant improvement in the continuity of the faults, as the red arrows indicate in Figure 15c,d.

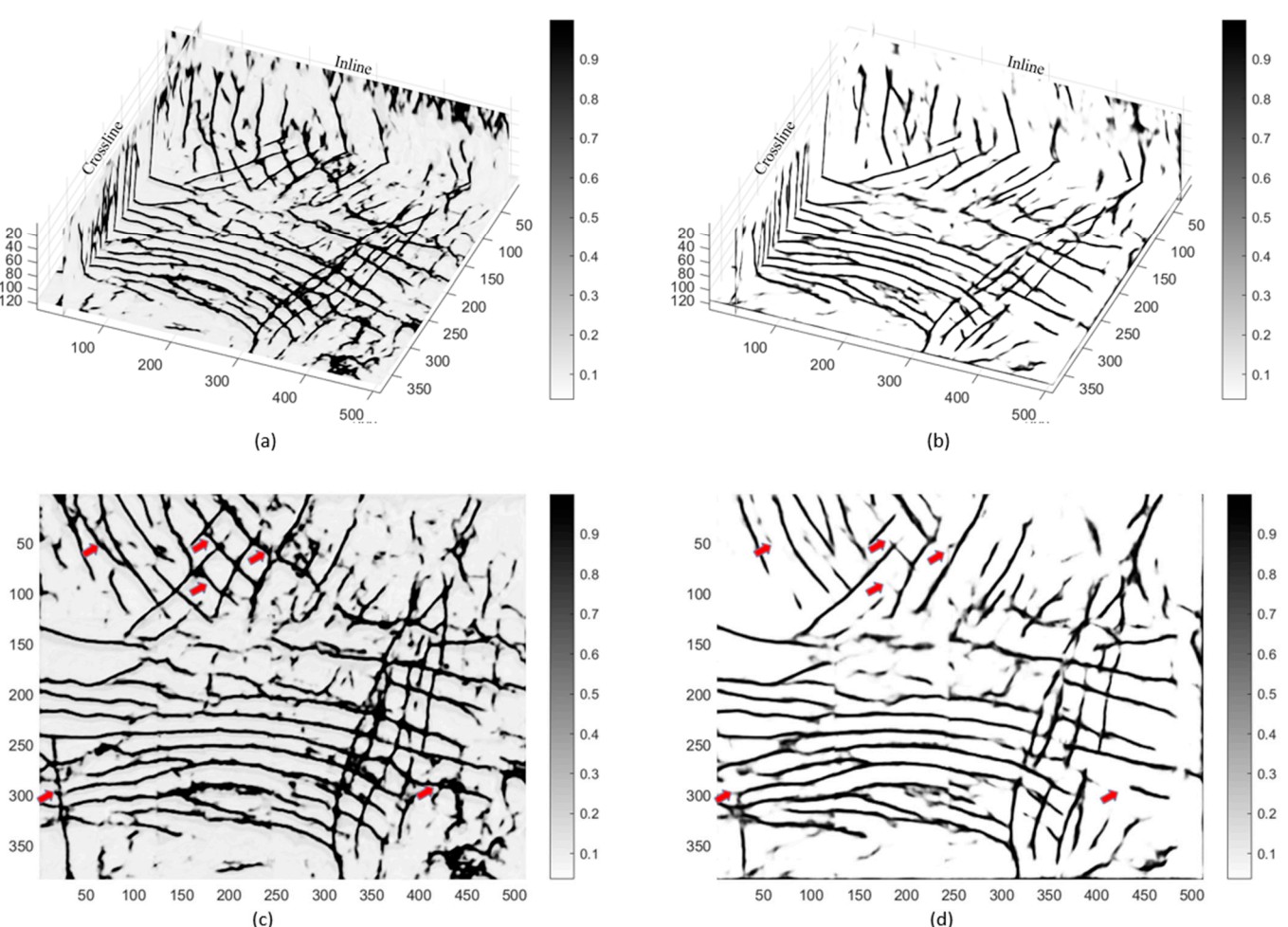

**Figure 15.** FaultSeg3D results. (**a**) Training with standardized data. (**b**) The result provided by Wu et al. [12] with original data. (**c**) The time slice in (**a**). (**d**) The time slice in (**b**).

## 6. Conclusions

The main conclusions of this paper are the following:

(1) To address with the shortcomings of existing convolution-based models, we propose 2.5D Transformer U-net for seismic fault detection tasks. The results on F3 field data show that the proposed method generates improved results compared with the 3D model FaultSeg3D. Our method predicts a more complete fault structure with better continuity.

(2) The proposed model consumes less GPU memory than the 3D model and a similar amount to GPU memory to 2D models. At the same time, the 2.5D method increases the continuity of the predicted faults even better than the 3D FaultSeg3D model.

(3) To reduce the discrepancy between synthetic training data and field data, we adopted a data standardization workflow. This is simple to operate and can be easily applied to other methods. The proposed data standardization is able to improve the model generalization and increase the stability of the neural network for fault detection based on synthetic data training.

The further reduce computational complexity, the use of Transformer directly for 3D fault detection to obtain more powerful prediction capability will be investigated in the future.

**Author Contributions:** Conceptualization, Z.T.; Data curation, W.W.; Investigation, B.W.; Methodology, Z.T.; Resources, D.M.; Software, W.W.; Validation, B.W.; Writing—original draft, Z.T.; Writing—review and editing, B.W. All authors have read and agreed to the published version of the manuscript.

**Funding:** This work was supported by Natural Science Basic Research Program of Shaanxi (Program No. 2023-JC-YB-269).

**Acknowledgments:** The authors would like to thank Xinming Wu for the synthetic seismic data and open-source code for comparison. The authors are also thankful to provider of Netherlands offshore F3 seismic data used in this paper.

**Conflicts of Interest:** The authors declare no conflict of interest.

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
