# Peer review of "Fault Detection via 2.5D Transformer U-Net with Seismic Data Pre-Processing"

_remotesensing, doi:10.3390/rs15041039_

Round 1

Reviewer 1 Report

In this manuscript, the authors propose a 2.5D Transformer U-net fault detection method. One of the relatively essential elements is the preprocessing of the training data and the real data so that they have similar amplitude distributions. From the results, the preprocessing does have improvement on the overfitting problem in network training. Overall, the manuscript is well organized and I have a few relatively minor queries and suggestions.

1, Line 154-157, The description of SW_MSA is a bit vague for me, I suggest to explain a bit more clearly why SW_MSA can exchange information between different windows.

2, Line 175-177, it would be better to demonstrate experiment results to prove the advantages of OPM over PBE.

3, In Figure 3, it would be better if the lower right square in the right figure was in the same color (orange) as the dashed line surrounding the lower right block in the left figure.

4. After step 1 of data preprocessing, the data have been normalized into 201 values. What I am more curious about is whether the choice of the number of values will affect the accuracy of the fault detection? If we aim to detect large scale faults, what is the appropriate number of values to choose? If the target is a small-scale fault, then what is the appropriate number of values to choose? Have the authors done any experiments accordingly?

5, Figure 10 compares the fault detection results of 2.5D and 2D networks. The results are logical because it is difficult to identify the fault in 2D slice when the fault strike is parallel to the direction of 2D slice. However, I feel that Fig. 10d and Fig. 10e may have mislabeled the direction of the 2D slice. As seen in Fig. 10e, the unidentified fault strike is parallel to the inline direction.

Reviewer 2 Report

Transformer has been widely used in the field of NLP and CV with impressive results. However, Transformer requires expensive computational resources, particularly when applied to 3D seismic data interpretation. The authors propose a 2.5D Transformer U-net and seismic data standardization workflow to detect seismic fault, and detection results indicate the proposed method is reliable. I recommend minor modifications and have following comments.

1. In line 12, line 23 and line 25, the format of words “on”, “standardization” and “faults” is different from the manuscript. There are still a few such details in manuscript. In line 388, please confirm the usage of the word “tomography” is correct.

2. Line 397-403. Generally speaking, 3D data provides better results than 2D. However, the authors said “When the number of channels exceeds a certain range, more channels may not more effective”. Why did the authors' experiment yield such results? Whether this is caused by the single channel label.

3. When you slice 2.5D training data, is there any spacing or overlapping? What is the number of 2.5D training data? Is the 2D training data also sliced from both inline and crossline directions?

4. Line 460-461, the authors point out that the training data is 2.5D and the network is 2D. I suggest the description here should be consistent with the title and manuscript (2.5D Transformer U-net).

5. Comparison with FaultSeg3D, does Transformer have advantage in terms of training or detection time consumption?

Reviewer 3 Report

The objective of this paper is analyzing the fault detection via 2.5D Transformer U-net by applying seismic data pre-processing. In particular, a data pre-processing workflow for data standardization is proposed and described in detail.

This is an interesting and well-structured paper, highlighting new aspects of the fault modelling determination. All necessary sections (Introduction, Network Structure, Data Processing, Experimental Results and Analysis, Conclusions) have been considered. Moreover, all Figures and Diagrams are consistent with the detailed analysis, provided in the manuscript, while the mathematical part (equations) is valid. However, some changes should be implemented, which will improve the paper. In particular:

Lines 9-26: Although the abstract has been properly structured, unnecessary details are contained. The abstract should be clear and concise, while the most significant processes/findings/conclusions should be highlighted. Please, modify the abstract by reducing its length.

Lines 30-33: The first paragraph of the “Introduction” section includes no references. Moreover, the role of faulting in the crust deformation is omitted, which is the key point for further analyses, such as hydrocarbon exploration. Typical papers, describing the crust deformation, which can be optionally cited, are the following: 1. McClusky, S., Balassanian, S., Barka, A., Demir, C., Ergintav, S., Georgiev, I., Gurkan, O., Hamburger, M., Hurst, K., Kahle, H., Kastens, K., Kekelidze, G., King, R., Kotzev, V., Lenk, O., Mahmoud, S., Mishin, A., Nadariya, M., Ouzounis, A., … Veis, G. (2000). Global Positioning System constraints on plate kinematics and dynamics in the eastern Mediterranean and Caucasus. Journal of Geophysical Research105(B3), 5695. https://doi.org/10.1029/1999JB900351, 2. Müller, M. D., Geiger, A., Kahle, H. G., Veis, G., Billiris, H., Paradissis, D., & Felekis, S. (2013). Velocity and deformation fields in the North Aegean domain, Greece, and implications for fault kinematics, derived from GPS data 1993-2009. Tectonophysics597–598, 34–49. https://doi.org/10.1016/j.tecto.2012.08.003, 3. Lazos, I., Papanikolaou, I., Sboras, S., Foumelis, M., & Pikridas, C. (2022). Geodetic Upper Crust Deformation Based on Primary GNSS and INSAR Data in the Strymon Basin, Northern Greece – Correlation with Active Faults. Applied Sciences 2022, Vol. 12, Page 939112(18), 9391. 4. Nyst, M., & Thatcher, W. (2004). New constraints on the active tectonic deformation of the Aegean. Journal of Geophysical Research B: Solid Earth109(11), 1–23. https://doi.org/10.1029/2003JB002830. Please, include the corresponding references.

Line 40: The correct citation style is [1-3]. Please, apply throughout the manuscript.

Lines 117-119: The last sentence of this paragraph should be included in the next paragraph. In particular the major aim of the paper should be described and then all steps/chapters should be described. Please, modify.

Line 159: The Figure 1 caption should include more details. Please, apply.

Line 317: The analysis of Figure 5 is quite poor. Please, increase the resolution.

Line 326:  The analysis of Figure 8 is similarly poor. Please, increase the resolution.

Line 410: I suggest renaming the “Comparison results on data standardization” into “Discussion”. This is more appropriate for this part of the manuscript.

Line 456: The “Conclusions” section should be modified. In the current form, it resembles an abstract rather than conclusions. This section should be comprehensive, while the major findings of the paper should be highlighted. Maybe, numbering of the conclusion remarks could be performed. The last sentence about the future perspectives can be maintained. Please, apply.
